# Balancing Robustness and Sensitivity using Feature Contrastive Learning

## Abstract

It is generally believed that robust training of extremely large networks is critical to their success in real-world applications. However, when taken to the extreme, methods that promote robustness can hurt the model's sensitivity to rare or under-represented patterns. In this paper, we discuss this trade-off between robustness and sensitivity by introducing two notions: *contextual feature utility* and *contextual feature sensitivity*. We propose Feature Contrastive Learning (FCL) that encourages the model to be more sensitive to the features that have higher contextual utility. Empirical results demonstrate that models trained with FCL achieve a better balance of robustness and sensitivity, leading to improved generalization in the presence of noise.

## 1 Introduction

Deep learning has shown unprecedented success in numerous domains (Krizhevsky et al., 2012; Szegedy et al., 2015; He et al., 2016; Hinton et al., 2012; Sutskever et al., 2014; Devlin et al., 2018), and robustness plays a key role in the success of neural networks. When we seek robustness, we are interested in having the same model prediction for small perturbations of the inputs. However such invariance to small perturbations can prove detrimental in some cases. As an extreme example, it is sometimes possible that a small perturbation to the input changes the human perceived class label, but the model is insensitive to this change (Tramèr et al., 2020). In this paper, we focus on balancing this tradeoff between robustness and sensitivity by developing a contrastive learning method that promotes the change in model prediction for certain perturbations, and inhibits the change for certain other perturbations. Note that we are only referring to non-adversarial robustness in this paper, i.e., we are not making any effort to improve robustness to carefully designed adversarial perturbations (Goodfellow et al., 2014).

To develop algorithms that balance robustness and sensitivity, we first formalize two measures: *utility* and *sensitivity*. Utility refers to the change in the loss function when we perturb a specific input feature. In other words, whether an input feature is useful for the model's prediction. Sensitivity, on the other hand, is the change in the learned embedding representation (before computing the loss) when we perturb a specific input feature. In contrast to classical feature selection approaches (Guyon & Elisseeff, 2003; Yu & Liu, 2004) that identify relevant and important features, our notions of sensitivity and utility are *context dependent* and change from one image to another. Our goal is to ensure that if an input feature has high utility, the model will also be sensitive to it, and if it has low utility then the model won't.

To explore and illustrate the notions of utility and sensitivity, we introduce a synthetic MNIST dataset, as shown in Figure 1. In the standard MNIST, the goal is to classify 10 digits based on their appearance. We modify it by adding a small random digit in the corner of some of the images and increasing the number of classes by five. For digits 5-9 we never change the class labels even in the presence of a corner digit, whereas digits 0-4 move to extended class labels 10-14 in the presence of any corner digit. The small corner digits can have high or low utility *depending on the context*. If the digit in the center is in 5-9 the corner digit has no bearing on the class, and will have low utility. However, if the digit in the center of the image is in 0-4, the presence of a corner digit is essential to determining the label, and thus has high utility. We would like to promote model sensitivity to the small corner digits when they are informative, in order to improve predictions, but demote it when they are not, in order to improve robustness.

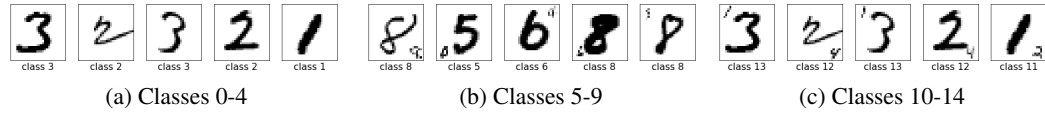

| (a) Classes 0-4 | (b) Classes 5-9 | (c) Classes 10-14 |

Figure 1: Synthetic MNIST data. We synthesize new images by adding a scaled down version of a random digit to a random corner. Images synthesized from digits 5-9 keep their label (Figure 1b) while images synthesized from digits 0-4 are considered to be of a different class (Figure 1c). In this setup corner pixels are informative only in a certain context.

**Feature attribution methods.** Our notions of utility and sensitivity are related to feature attribution methods. Given an instance $x$ and a model $f$, feature based explanation aims to attribute the prediction of $f(x)$ to each feature. There have been two different approaches to understand the role of features. In the former, we compute the derivative of $f(x)$ with respect to each feature, which is similar to the sensitivity measure proposed in this paper (Shrikumar et al., 2017; Smilkov et al., 2017; Simonyan et al., 2013; Sundararajan et al., 2016). The latter methods measure the importance by removing a feature or comparing it with a reference point (Samek et al., 2016; Fong & Vedaldi, 2017; Dabkowski & Gal, 2017; Ancona et al., 2018; Yeh et al., 2019; Zeiler & Fergus, 2014; Zintgraf et al., 2017). For example, the idea of prediction difference analysis is to study the regions in the input image that provide the best evidence for a specific class or object by studying how the prediction changes in the absence of a specific feature. While many of the existing methods look at the interpretability of the model predictions, our work proposes loss functions in the training stage to adjust the sensitivity according to their utility in a context-dependent manner.

**Robustness.** It is widely believed that imposing robustness constraints or regularization to neural networks can improve their performance. Taking the idea of robustness to the extreme, adversarial training algorithms aim to make neural networks robust to any perturbation within an $\epsilon$-ball (Goodfellow et al., 2014; Madry et al., 2017). The certified defense methods pose an even stronger constraint in training, i.e., the improved robustness has to be verifiable (Wong & Kolter, 2018; Zhang et al., 2019b). Despite being successful in boosting accuracy under adversarial attacks, they come at the cost of significantly degrading clean accuracy (Madry et al., 2017; Zhang et al., 2019a; Wang & Zhang, 2019). Several theoretical works have demonstrated that a trade-off between adversarial robustness and generalization exists (Tsipras et al., 2018; Schmidt et al., 2018). Recent papers (Laugros et al., 2019; Gulshad et al., 2020) also discuss the particular relationship between adversarial robustness and natural perturbation robustness, and find that they are usually poorly correlated. For example, Laugros et al. (2019) shows models trained for adversarial robustness that are not more robust than standard models on common perturbation benchmarks and the converse holds as well. (Gulshad et al., 2020) also found a similar trend while natural robustness can commonly improve adversarial robustness slightly. While adversarial robustness is important in its own way, this paper mainly focus on natural perturbation robustness. In fact, our goal of "making models sensitive to important features" implies that the model should not be adversarially robust on high utility features.

With the goal of improving generalization instead of adversarial robustness, several other works enforce a weaker notion of robustness. A simple approach is to add Gaussian noise to the input features in the training phase. Lopes et al. (Lopes et al., 2019) recently showed that Gaussian data augmentation with randomly chosen patches can improve generalization. (Xie et al., 2020) showed that adversarial training with a dual batch normalization approach can improve the performance of neural networks. It is worth noting a closely related work (Kim et al., 2020), which also employs contrastive learning for robustness (See Section 3 for details); however, it differs in three main aspects: a) their paper focuses on adversarial robustness while ours focuses on robustness to natural perturbations b) their contrastive learning always suppresses the distance between the original and an adversarially perturbed image while our proposal encourages to differ for high-utility perturbation pairs and suppress for the low-utility pairs c) their perturbation is based on an unsupervised loss, while we rely on class labels to identify low and high utility features with respect to the classification task.

In summary, all the previous works in robust training aim to make the model insensitive to perturbation, while we argue that a good model (with better generalization performance) should be robust to

unimportant features while being sensitive to important features. A recent paper in the adversarial robustness community also pointed out this issue (Tramèr et al., 2020), where they showed that existing adversarial training methods tend to make models *overly robust* to certain perturbations that the models should be sensitive to. However, they did not provide any solution to the problem of how to balance robustness and sensitivity.

**Main contributions**

- We propose *contextual sensitivity* and *contextual utility* metrics that allows to measure and identify high utility features and their associated model sensitivity.

- We propose Feature Contrastive Learning (FCL) that promotes model sensitivity to perturbations of high utility features, and inhibits model sensitivity to perturbations of low utility features.

- We demonstrate practical utility of our approach on a synthetic dataset as well as real-world CIFAR-10, CIFAR-100, and ImageNet datasets (with noise injection and corruption patterns).

## 2 ROBUSTNESS AND SENSITIVITY

### 2.1 BACKGROUND AND NOTATION

Before we formally define contextual utility and contextual sensitivity, we consider a simple scenario for motivation. Consider binary classification of images with 0/1 loss. For a specific input image, changing a pixel a little could lead to a change in the model's prediction (i.e. best guess) or the label. When the prediction changes, we say that the model is contextually sensitive to this pixel. Sensitivity is thus independent of the label. The change in the pixel may or may not affect the loss, since the label may change as well. We measure the contextual utility of the pixel with respect to its effect on the loss on the specific image. While utility and sensitivity are related, neither implies the other: Binary loss does not change when both the prediction and the label change. In this case, the model is sensitive to the pixel, but the pixel utility (for the specific image) is zero. On the other hand, when only the label changes, the model is not sensitive to the pixel, but the pixel has high utility. Ideally, we would like the model to be sensitive to pixels that have high utility. For the same pixel in the same image, both utility and sensitivity depend on the model parameters and hence evolve along with the model in the training stage.

We generalize these concepts to multi-class classification with loss functions that are differentiable with respect to input features. We also relate sensitivity to the model's probability distribution over the classes - rather than focusing on its best guess. We can define it with respect to change in the probability distribution. We can also define it relative to a specific model architecture. For example, sensitivity can be defined with respect to change in the logits or in the embedding representation at a specific layer (typically the layer before the logits) in deep neural networks. We highlight one choice in the formal definition later in this section and use it in all our experiments.

**Multiclass classification**   Let us consider a classification setting with $L$ classes. We are given a finite set of $n$ training samples $\mathcal{S} = \{(x_1, y_1), \ldots, (x_n, y_n)\}$, where $x_i \in \mathcal{X}$ and $y_i \in \mathcal{Y}$. Here $\mathcal{X}$ and $\mathcal{Y}$ denote the instance and output spaces with dimensions $D$ and $L$ respectively. The output vector $y_i$ is treated as the 1-hot encoding of the class labels. Let $f : \mathcal{X} \to \mathbb{R}^L$ be the function that maps the input vector to one of the $L$ classes. Accordingly, given a loss function $\ell : \{0, 1\}^L \times \mathbb{R}^L \to \mathbb{R}_+$, our goal is to find the parameters $w^*$ that minimize the expected loss:

$$w^* = \arg\min_w \mathbb{E}_{y \sim \mathcal{Y}, x \sim \mathcal{X}} \ell(y, f(x; w)).$$

In this work, we consider the cross entropy loss function $\ell(y, f_w(x)) = \sum_c \mathbf{1}_{y=c} \log f(x; w)_c$, but our formulation is not restricted to this loss. The model $f(x) : \mathcal{X} \to \mathbb{R}^L$ can be seen as the composition of an embedding function $\phi : \mathcal{X} \to \mathbb{R}^E$ that maps an input to an $E$-dimensional feature, and a discriminator function $h : \mathbb{R}^E \to \mathbb{R}^L$ that maps a learned embedding to an output. In other words, $f(x; w) = (h \circ \phi)(x; w_\phi, w_h)$ and $w = \{w_\phi, w_h\}$.

Given a finite training set $\mathcal{S}$, we minimize the following empirical risk to learn the parameters:

$$w^* = \arg\min_w \frac{1}{N} \sum_{(x_i, y_i) \sim \mathcal{S}} \ell\left(y_i, f(x_i; w)\right).$$

## 2.2 CONTEXTUAL FEATURE UTILITY

**Definition 1** (*Contextual feature utility*). Given a model $f : \mathcal{X} \to \mathbb{R}^L$ and a loss function $\ell : \{0, 1\}^L \times \mathbb{R}^L \to \mathbb{R}_+$, the *contextual utility vector* associated with a training sample $(x_i, y_i) \in \mathcal{S}$, is given by:

$$u_i = \left| \frac{\partial \ell(y_i, f(x_i; w))}{\partial x_i} \right|, \tag{1}$$

where $i$ denotes the index of a specific training sample. The element $u_{ij}$ in the vector $u_i$ denotes the utility along the relevant input dimension $j$. Note that the contextual feature utility vector is nothing but the absolute value of Jacobian of the loss function with respect to the input vector, and the Jacobian has been shown to be closely related to stability of the network (Jakubovitz & Giryes, 2018).

The contextual utility $u_{ij}$ denotes the change in the loss function $\ell$ with respect to perturbation of the input sample $x_i$ along the dimension $j$. A perturbation of the high utility feature leads to a larger change in loss compared to the perturbation of the low utility feature. Please note that this utility function is context sensitive, i.e., the dimension having high utility for one training sample may have low utility for another sample.

## 2.3 CONTEXTUAL FEATURE SENSITIVITY

**Definition 2** (*Contextual feature sensitivity*). Given an embedding function $\phi : \mathcal{X} \to \mathbb{R}^L$, the sensitivity $s_{ij}$ associated with a training sample $(x_i, y_i) \in \mathcal{S}$ is as follows:

$$s_{ij} = \left\| \frac{\partial \phi(x_i, w_\phi)}{\partial x_{ij}} \right\| \tag{2}$$

Sensitivity is nothing but the norm of the Jacobian of the embedding function with respect to the input. The notion of sensitivity captures how the features corresponding to an input $x_i$ change for small perturbations of the input along dimension $j$. Similar to utility, the sensitivity is also context dependent and changes from one training sample to another. Note that the sensitivity could also be defined on the embeddings from intermediate layers, as well as the final output space. Driven by the empirical success of other stability training (Zheng et al., 2016) and contrastive learning methods (Chen et al., 2020), we choose to develop contrastive loss functions in the embedding space defined by the penultimate layer of the network. In contrast to the feature utility vector that depends on the true class labels, the feature sensitivity is independent of the class labels. Please see Appendix A for a more detailed discussion of the relationship between contextual feature utility and sensitivity.

## 3 FEATURE CONTRASTIVE LEARNING

Our goal is to learn an embedding function $\phi : \mathcal{X} \to \mathbb{R}^L$ that is more *sensitive* to the features with higher contextual utility than the ones with lower contextual utility. That is, we want embeddings of examples perturbed along low utility dimensions to remain close to the original embeddings, and embeddings of examples perturbed along high utility dimensions to be far. Our formulation utilizes the contextual utility and sensitivity and the interplay between them. The utility is used for selecting the features, and the associated sensitivity values are adjusted by applying the contrastive loss.

We now describe a method to achieve this goal, using a contrastive loss on embeddings, derived from utility-aware perturbations. In typical contrastive learning methods (Chen et al., 2020), positive and negative pairs are generated using data augmentations of the inputs, and the contrastive loss function minimizes the distance between embeddings from positive pairs, and maximizes the distances between embeddings from negative pairs. We follow the same path, but use contextual utility to define the positive and negative sets.

---

**Algorithm 1** FCL algorithm

---

Initialize model $f : \mathcal{X} \to \mathbb{R}^L$ with parameters $w_0$
**for** Sample minibatch $S = [(x_1, y_1), ...(x_n, y_n)]$ from $\mathcal{S}$ **do**
$\quad \forall_i \; u_i = \left| \frac{\partial \ell(y_i, f(x_i; w))}{\partial x_i} \right|,$
$\quad$ **for** $i \in \{1, ...n\}$ **do**
$\qquad z_i = \phi(x_i, w_\phi)$
$\qquad z_i^+ = \phi(x_i + \epsilon(\text{BOTTOM}_k(u_i)), w_\phi)$
$\qquad z_i^- = \phi(x_i + \epsilon(\text{TOP}_k(u_i)), w_\phi)$
$\quad$ **end for**
$\quad$ Let $\ell_{\text{FCL}} = \sum_i \ell_{\text{xe}}^i$ or $\ell_{\text{FCL}} = \sum_i \ell_{\text{margin}}^i$
$\quad$ Update model parameters: $w_{t+1} \leftarrow w_t - \eta \frac{\partial \ell + \lambda \ell_{\text{FCL}}}{\partial w}.$
**end for**

---

**Definition 3** (*Utility-aware perturbations*). Let $\text{TOP}_k(v)$ and $\text{BOTTOM}_k(v)$ denote the largest and smallest $k$ indices of vector $v$ (ties resolved arbitrarily), respectively. Let $\epsilon(\mathcal{S})$ denote perturbation vectors of dimension $D$ such that

$$\epsilon(\mathcal{S})_i \begin{cases} \sim \mathcal{N}(0, \sigma^2), & \text{if } i \in \mathcal{S} \\ = 0, & \text{otherwise} \end{cases} \tag{3}$$

Using the utility vector $u_i$ for a training sample $x_i$, we refer to $\epsilon(\text{TOP}_k(u_i))$ as the high-utility perturbation, and $\epsilon(\text{BOTTOM}_k(u_i))$ as the low-utility perturbation.

For simplicity, let us use $z = \phi(x, w_\phi)$ to denote the embedding associated with the input $x$. In order to increase the sensitivity along high utility features, we add a high-utility perturbation, $z_i^- = \phi(x_i + \epsilon(\text{TOP}_k(u_i)), w_\phi)$. Similarly, in order to decrease the sensitivity along low utility features, we add a low-utility perturbation, $z_i^+ = \phi(x_i + \epsilon(\text{BOTTOM}_k(u_i)), w_\phi)$. Our key idea is to treat $(z_i, z_i^+)$ as a positive pair, and $(z_i, z_i^-)$ as a negative pair in a contrastive loss. In other words, we want to do deep metric learning such that the high-utility perturbations lead to distant points and low-utility perturbations lead to nearby points in the embedding space.

For a given sample $x_i$, we have a single positive pair $\mathcal{P}_i = \{(z_i, z_i^+)\}$ and a set of negative pairs $\mathcal{N}_i$, which consists of $(z_i, z_i^-)$ and $(z_i, z_j)$ where $j \neq i$.

We can now adapt any contrastive loss from the literature to our positive and negative pairs. We define two versions to show the flexibility, but focus on one of them for all our experiments.

**Definition 4** (*Feature Contrastive Loss*). Given the positive pair $\mathcal{P}_i$ and the set of negative pairs $\mathcal{N}_i$ for a sample $x_i$, we define the two variants for the Feature Contrastive Loss ($\ell_{\text{FCL}}$) as follows:

$$\ell_{\text{margin}}^i = \text{dist}(z_i, z_i^+)^2 + \sum_{(z_i, z_j) \in \mathcal{N}_i} \max(0, \gamma - \text{dist}(z_i, z_j))^2, \tag{4}$$

$$\ell_{\text{xe}}^i = -\log \frac{e^{(1 - \text{dist}(z_i, z_i^+))/\tau}}{e^{(1 - \text{dist}(z_i, z_i^+))/\tau} + \sum_{(z_i, z_j) \in \mathcal{N}_i} e^{(1 - \text{dist}(z_i, z_j))/\tau}}, \tag{5}$$

where the first loss is the contrastive loss based on margin $\gamma$, similar to the one proposed in (Chopra et al., 2005), and the second variant is based on a recent contrastive learning method (Chen et al., 2020). Here $\text{dist}(a, b)$ denotes the cosine distance given by $1 - \frac{a^T b}{|a||b|}$. Once the positive and negative pairs are generated using our formulation, we can use different variants of contrastive loss functions.

Equation 4 and Equation 5 solve the same problem with different approaches. Equation 4 strictly minimizes the distance between the $z_i$ and $z_i^+$ and encourages a margin of at least $\gamma$ between $z_i$ and $z_j \in \mathcal{N}_i$. Equation 5, on the other hand, applies a softer contrast between the rankings of $\text{dist}(z_i, z_i^+)$ and $\text{dist}(z_i, z_j \in \mathcal{N}_i)$ similar to the softmax cross entropy loss. We choose this version for all our experiments, after observing comparable performance on the synthetic MNIST experiments described in Section 4.1.

Algorithm 1 describes the details of FCL algorithm. It's important to note that during early stages of training, the utility is likely to fluctuate and be very noisy. Imposing sensitivity constraints based on the early stage utility can be detrimental. We therefore use a warm-up schedule. We keep $\lambda = 0$ until a certain number of training epochs and then switch it to a fixed positive value for the rest of the training.

# 4 EXPERIMENTS

## 4.1 SYNTHETIC MNIST CLASSIFICATION

To illustrate how FCL can balance robustness and sensitivity, we generate a synthetic dataset based on MNIST digits (LeCun et al., 1989). We set up the task so that some patterns are not useful most of the time, but are very informative in a certain context, which occurs rarely. We show that by using FCL, our models i) maintain sensitivity *in the right context* and ii) become more robust by suppressing uninformative features.

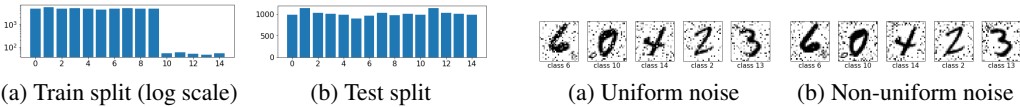

(a) Train split (log scale)     (b) Test split       (a) Uniform noise     (b) Non-uniform noise

Figure 2: Class distribution. Train split is highly unbalanced with classes 10-14 appearing rarely.

Figure 3: Two types of random noise, used to evaluate notions of robustness.

**Data generation** The original MNIST images consist of a single digit centered over a uniform background, the corners of the image are empty in almost all examples, as seen in Figure 1a. We synthesize new images by adding a scaled down version of a random digit to a random corner, as seen in Figures 1b and 1c. The images synthesized from digits 0-4 are considered to be new classes, classes 10-14 respectively. Examples are shown in Figure 1c. In contrast, images synthesized from digits 5-9 do not change the class label, as shown in Figure 1b. For the new images, the small digits in the corners are uninformative except in a certain context. If the digit in the center is in 5-9 the corner digit has no bearing on the class, but if the digit in the center of the image is in 0-4, the presence of a corner digit is essential to determining if the image should be labeled as 0-4 or as 10-14.

**Experiment** We generate a training set in which the new classes, classes 10-14 are very rare (see Figure 2a), appearing with a ratio of approximately $1/100$ compared to classes 0-9. Classes 0-9 have approximately 5000 examples each, while classes 10-14 have approximately 50 each. The challenge for models trained with this data is that the small digits in the corners are going to be completely uninformative 100 out of 101 times they appear. To emphasize the importance of learning the rare classes, our test (and validation) sets have a balanced distribution over all classes (Figure 2b). The balanced test set is labeled 'BAL'. In total, we have roughly 50k training examples, 15k validation examples and 15k test examples. The validation set has a distribution similar to the test set's and was used to tune hyper-parameters.

To demonstrate that FCL increases robustness to noise, we also prepare two noisy versions of the balanced test set. In both of these test sets we replace $15\%$ of the pixels, with a uniformly chosen random gray level. For the uniform noise test set (shown in Figure 3a) the location of the noisy pixels is chosen uniformly. We label this set 'BAL+UN'. For the *non-uniform noise* test set, 'BAL+NUN', (shown in Figure 3b) the probability of a pixel being replaced with noise is inversely proportional to its sample standard deviation (over training images). The intuition is that in this set, noise will be concentrated in less "informative" pixels.

We train a LeNet-like convolutional neural network (CNN) (LeCun et al., 1989). The network is trained for 20 epochs using the Adam optimizer (Kingma & Ba, 2014), with an initial learning rate of 0.01 and exponential decay at a rate of 0.89 per epoch. FCL is turned on after 2 epochs with a linear warmup of 2 epochs. We set $k = 256, \lambda = 0.001, \tau = 0.1$ and $\sigma = 0.5$ (image values are in $[0, 1]$). These values were determined empirically using the validation set. In later stages of training



Figure 4: Visualization of utility averaged over classes (modulo 10) within one batch. Each of the ten panels shows the average image on the left, and the average utility on the right. We can see that the corner pixels have high utility *only in a certain context*. When the central digit is 0-4, the corner pixels are important, since they can flip the class, but when the central digit is 5-9 they are not.

| Dataset | Method | BAL | BAL+UN | BAL+NUN |
|---------|--------|-----|--------|---------|
| Synthetic MNIST | XE | $0.9250 \pm 0.0088$ | $0.4123 \pm 0.1176$ | $0.5473 \pm 0.0703$ |
| Synthetic MNIST | $FCL_{xe}$ | $0.9207 \pm 0.0129$ | $\textbf{0.6384} \pm \textbf{0.0530}$ | $\textbf{0.6896} \pm \textbf{0.0349}$ |

Table 1: Average accuracy and standard deviation over 10 runs, on the synthetic MNIST test sets.

the utility values become very small. To avoid numerical issues we drop high utility perturbations if the max utility value is smaller than $\epsilon = 10^{-12}$. Each experiment is repeated 10 times.

**Results** The mean accuracy and the standard deviation are shown in Table 1. Results on the noisy test sets 'BAL+NU' and 'BAL+NUN' show that using FCL can significantly improve robustness to noise while maintaining sensitivity. Figure 4 illustrates the context dependent utility of the small digits in the corners of the image. This is the signal used by FCL to emphasize contextual sensitivity. Note that the models don't see any noisy images in training, they can however learn which pixels are less informative in certain contexts and suppress reliance on those.

## 4.2 Larger-scale experiments

To evaluate FCL's performance on general tasks, we conducted experiments on public large-scale image datasets (CIFAR-10, CIFAR-100, ImageNet) with synthetic noise injection similar to Section 4.1, and with the 19 predefined corruption patterns from (Hendrycks & Dietterich, 2019) – called CIFAR-10-C, CIFAR-100-C and ImageNet-C. We show that FCL can significantly improve robustness to these noise patterns, with minimal, if any, sacrifice in accuracy.

**Baselines** Apart from the standard cross-entropy baseline 'XE', we consider three other baselines 'XE+Gaussian', 'CL+Gaussian' and 'Patch Gaussian+XE'. In 'XE+Gaussian', all the image pixels are perturbed by Gaussian noise, and an additional cross-entropy term (weighted by a scalar $\lambda$) is applied to perturbed versions of the image, keeping the original label. In 'CL+Gaussian', we add a contrastive loss similar in form to $\ell_{xe}$ (Equation 5) to the original cross-entropy classification loss. We use the same weight $\lambda$ as in $FCL_{xe}$ but with a random Gaussian perturbed image as the positive pair instead of the utility-dependent perturbation. 'Patch Gaussian', recently proposed by (Lopes et al., 2019) is a data augmentation technique. An augmentation is generated by adding a patch of random Gaussian noise to a random position in the image. This technique achieved state-of-the-art performance on CIFAR-10-C. In 'XE+Gaussian' the perturbation is applied to all features, in 'Patch Gaussian+XE' it is applied to a subset of the pixels, chosen at random, while FCL applies perturbations to a subset of pixels based on *contextual utility*. Note that since Patch Gaussian is purely a data augmentation technique, it can easily be combined with FCL, as we do in 'Patch Gaussian+$FCL_{xe}$'.

**Model and hyperparameters** ResNet-56 was used for CIFAR experiments and ResNet-v2-50 for the ImageNet experiment. We used the same common hyper-parameters such as learning rate schedule and the use of SGD momentum optimizer (0.9 momentum) across all experiments. Details on hyper-parameters, learning rate schedules and optimization can be found in Appendix B. Models are trained for 450 epochs and contrastive learning losses (FCL and CL+Gaussian) are applied after 300 epochs (CIFAR) or 60 epochs (ImageNet). We kept all standard CIFAR/ImageNet data

| Dataset | Method | Clean | UN | NUN |
|---|---|---|---|---|
| Noisy CIFAR-10 | XE | $0.9389 \pm 0.0014$ | $0.1317 \pm 0.0135$ | $0.1256 \pm 0.0089$ |
| | XE+Gaussian | $0.9375 \pm 0.0016$ | $0.3409 \pm 0.0580$ | $0.3175 \pm 0.0532$ |
| | CL+Gaussian | $0.9362 \pm 0.0009$ | $0.2646 \pm 0.0165$ | $0.2464 \pm 0.0159$ |
| | $FCL_{xe}$ | $0.9375 \pm 0.0010$ | $\mathbf{0.3749 \pm 0.0293}$ | $\mathbf{0.3432 \pm 0.0231}$ |
| | Patch Gaussian+XE | $0.9334 \pm 0.0035$ | $0.7842 \pm 0.0087$ | $0.7669 \pm 0.0086$ |
| | Patch Gaussian+$FCL_{xe}$ | $0.9354 \pm 0.0023$ | $\mathbf{0.8210 \pm 0.0013}$ | $\mathbf{0.8066 \pm 0.0033}$ |
| Noisy CIFAR-100 | XE | $0.7323 \pm 0.0052$ | $0.0366 \pm 0.0078$ | $0.0356 \pm 0.0084$ |
| | XE+Gaussian | $0.7297 \pm 0.0057$ | $0.0806 \pm 0.0187$ | $0.0763 \pm 0.0162$ |
| | CL+Gaussian | $0.7294 \pm 0.0022$ | $0.0668 \pm 0.0122$ | $0.0640 \pm 0.0134$ |
| | $FCL_{xe}$ | $0.7252 \pm 0.0076$ | $\mathbf{0.1477 \pm 0.0227}$ | $\mathbf{0.1007 \pm 0.0160}$ |
| | Patch Gaussian+XE | $0.7315 \pm 0.0028$ | $0.0385 \pm 0.0102$ | $0.0377 \pm 0.0091$ |
| | Patch Gaussian+$FCL_{xe}$ | $0.7254 \pm 0.0045$ | $\mathbf{0.1590 \pm 0.0200}$ | $\mathbf{0.1033 \pm 0.0174}$ |

Table 2: Average accuracy and standard deviation over 5 runs on the noisy CIFAR test sets. *Gaussian* is adding the Gaussian noise uniformly across all feature dimensions and *Patch Gaussian* data augmentation is from (Lopes et al., 2019).

augmentations (random cropping and flipping) across all runs and added Patch Gaussian before or after the standard data augmentation as in (Lopes et al., 2019) when specified. For both Gaussian noise baselines, we swept $\sigma = [0.1, 0.3, 0.5]$ to choose the best performing parameter. For the Patch Gaussian, we used the code and the recommended configurations from (Lopes et al., 2019) – CIFAR-10: patch size=25, $\sigma = 0.1$, ImageNet: patch size$\leq 250$, $\sigma = 1.0$. Since CIFAR-100 parameters were not provided from the paper, we started from CIFAR-10 parameters and made our best effort to sweep the parameters (patch size=[15...25], $\sigma = [0.01...0.1]$). For contrastive learning methods, we swept $\lambda = [0.0001...0.0004]$ and $\tau = [2, 1, 0.5, 0.1]$. For FCL, we swept $k = [256, 512, 1024, 2048]$ and $\sigma_\epsilon = [0.1, 0.3, 0.5]$. We repeated all experiments 5 times.

### 4.2.1 NOISY CIFAR IMAGES

We follow the same protocol described in the synthetic MNIST experiment (Section 4.1) to generate uniform noise 'UN' and non-uniform noise 'NUN' test sets for CIFAR-10 and CIFAR-100. Table 2 demonstrates that $FCL_{xe}$ outperforms all baseline models with or without the PG data augmentation.

**Noisy CIFAR-10**   We can observe that Gaussian perturbation does improve performance in both UN and NUN (XE vs. XE+Gaussian or XE vs. CL+Gaussian); however, FCL's selective perturbation on the high contextual utility features obtains a better improvement in all cases (both Gaussian baselines vs. $FCL_{xe}$). When combined with the PG data augmentation, the gap between the clean accuracy versus 'UN' or 'NUN' narrows (0.93 vs 0.82). The combined version (Patch Gaussian+$FCL_{xe}$) achieves the best noisy CIFAR-10 performance (on 'UN' and 'NUN'), without hurting the clean accuracy .

**Noisy CIFAR-100**   Without the PG data augmentation, the pattern is similar to the case above; however gaps between XE, Gaussian baselines and $FCL_{xe}$ are wider suggesting that FCL gives more benefit when the number of classes is larger. PG did not work well in the 100 class setting, even with extensive tuning (including the recommended configurations from (Lopes et al., 2019)). The combination (Patch Gaussian+$FCL_{xe}$) achieves the best performance on 'UN' and 'NUN'.

### 4.2.2 CIFAR-10-C, CIFAR-100-C AND IMAGENET-C

We conducted a similar experiment on the public benchmark set of corrupted images (Hendrycks & Dietterich, 2019). This benchmark set evaluates robustness to natural perturbations of a prediction model by applying 19 common corruption patterns to CIFAR and ImageNet images. Table 3 shows the averaged accuracy on all corruption patterns, as well as the averages from each corruption pattern group. The full results are provided in Appendix C.

| Dataset | Method | All average | Noise | Blur | Weather | Digital |
|---|---|---|---|---|---|---|
| CIFAR-10-C | XE | 0.7137 ±0.0038 | 0.4967 | 0.6833 | 0.8309 | 0.7537 |
| | XE+Gaussian | 0.7379 ±0.0089 | 0.5800 | **0.6967** | **0.8389** | 0.7572 |
| | CL+Gaussian | 0.7253 ±0.0028 | 0.5446 | 0.6939 | 0.8312 | 0.7530 |
| | FCL$_{xe}$ | **0.7446 ±0.0055** | **0.6416** | 0.6886 | 0.8338 | 0.7530 |
| | Patch Gaussian+XE | 0.8311 ±0.0027 | 0.8951 | 0.7625 | 0.8540 | 0.8021 |
| | Patch Gaussian+FCL$_{xe}$ | 0.8319 ±0.0029 | **0.8993** | **0.7639** | 0.8536 | 0.8000 |
| CIFAR-100-C | XE | 0.4428 ±0.0038 | 0.2113 | 0.4323 | 0.5527 | 0.4855 |
| | XE+Gaussian | 0.4512 ±0.0067 | 0.2502 | 0.4308 | 0.5524 | 0.4848 |
| | CL+Gaussian | 0.4480 ±0.0057 | 0.2350 | 0.4350 | 0.5514 | 0.4865 |
| | FCL$_{xe}$ | **0.4706 ±0.0031** | **0.3528** | 0.4355 | 0.5467 | 0.4847 |
| | Patch Gaussian+XE | 0.4448 ±0.0030 | 0.2198 | 0.4344 | 0.5483 | 0.4896 |
| | Patch Gaussian+FCL$_{xe}$ | **0.4742 ±0.0054** | **0.3699** | 0.4353 | 0.5490 | 0.4851 |
| ImageNet-C | XE | 0.3406 ±0.0007 | 0.2615 | 0.2816 | 0.4214 | 0.3783 |
| | XE+Gaussian | 0.3414 ±0.0012 | 0.2623 | 0.2829 | **0.4224** | 0.3783 |
| | CL+Gaussian | 0.3418 ±0.0016 | 0.2658 | 0.2824 | 0.4223 | 0.3778 |
| | FCL$_{xe}$ | **0.3437 ±0.0022** | **0.2696** | 0.2850 | 0.4188 | **0.3827** |
| | Patch Gaussian+XE | 0.3625 ±0.0023 | 0.3053 | **0.3041** | 0.4300 | 0.3964 |
| | Patch Gaussian+FCL$_{xe}$ | **0.3634 ±0.0045** | **0.3077** | 0.3034 | 0.4308 | **0.3976** |

Table 3: Image classification accuracy on the corrupted CIFAR and ImageNet image sets (Hendrycks & Dietterich, 2019). The average column is based on all 19 corruption patterns and the other columns averages each corruption pattern group. The full table is in Appendix C.

**CIFAR-10-C** The pattern is similar to the noisy CIFAR-10. Adding the Gaussian perturbation improves the average performance, particularly in the noise-corruption pattern group and the blurring group. FCL$_{xe}$ works much better than Gaussian baselines providing an additional large improvement in the noise group. PG augmentation works really well for this task, particularly in the noise and blur groups (currently it is a state-of-the-art). Nevertheless, adding FCL$_{xe}$ can still add some value to some of the patterns. Appendix C shows FCL performs better for impulse noise and zoom blur patterns, while the vanilla PG performs better in fog and pixelization.

**CIFAR-100-C** Without PG, the improvement of FCL$_{xe}$ over the other baselines is even larger than for CIFAR-10-C, with a drastic improvement on the noisy corruption group. Similar to Noisy CIFAR-100, PG did not perform well in this setting, while PG+FCL$_{xe}$ still was able to perform well, achieving even better accuracy than without PG.

**ImageNet-C** With or without PG, FCL$_{xe}$ outperforms the baselines with the large improvements on 'digital', and 'noise' corruption patterns. Among individual patterns (reported in Appendix C), FCL performs particularly well on the 'shot' corruption pattern.

## 5 SUMMARY

In this paper, we propose Feature Contractive Learning (FCL), a novel approach to balance robustness and sensitivity in deep neural network training. Unlike previous work that only enforces robustness, FCL aims to promote model sensitivity to perturbations of high utility features, and inhibit model sensitivity to perturbations of low utility features. The performance of FCL is validated on both synthetic and real image classification datasets.

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

## A  CONNECTION BETWEEN CONTEXTUAL FEATURE UTILITY AND SENSITIVITY

Consider a classification task with cross entropy loss, and let $f()$ be the output of the network after applying a softmax. In this setting, the loss is minus log probability of the correct label. The contextual utility of $f()$ for a given feature is defined by

$$u = \left| \frac{\partial \ell(y, f(x; w))}{\partial x} \right| = \left| \frac{\partial \log[f(x; w)_y]}{\partial x} \right| = \frac{1}{f(x; w)_y} \left| \frac{\partial f(x; w)_y}{\partial x} \right|. \tag{6}$$

Also recall that the contextual sensitivity of $f()$ for a given feature is given by

$$s = \left\| \frac{\partial f(x; w)}{\partial x} \right\| = \sqrt{\frac{\partial f(x; w)_y^2}{\partial x} + \sum_{c \neq y} \frac{\partial f(x; w)_c^2}{\partial x}}. \tag{7}$$

We can see that the contextual feature utility is a product of two terms. The first is the reciprocal of the networks' prediction for the correct class, and the second is a sensitivity-like term specific to the correct class. When the network's prediction is correct the utility is proportional to the ground truth class's sensitivity. If changing the feature will not affect the correct prediction it doesn't have much utility and vice versa. On the other hand, when the network makes a mistake, the utility will be large regardless of the ground truth class's sensitivity. Our algorithm takes advantage of this behavior to promote robustness and maintain contextual sensitivity.

## B  EXPERIMENTAL SETUP

**Architecture**  For CIFAR experiments, we used a ResNet-56 architecture, with the following configuration for each ResNet block ($n_{\text{layer}}$, $n_{\text{filter}}$, stride): [(9, 16, 1), (9, 32, 2), (9, 64, 2)].

For ImageNet experiments, we used a ResNet-v2-50 architecture, with the following configuration for the ResNet block ($n_{\text{layer}}$, $n_{\text{filter}}$, stride): [(3, 64, 1), (4, 128, 2), (6, 256, 2), (3, 512, 2)].

**Optimization**  For CIFAR, we used SGD momentum optimizer (Nesterov=True, momentum=0.9) with a linear learning rate ramp up for 15 epochs (peaked at 1.0) and a step-wise decay of factor 10 at epochs 200, 300, and 400. In total, we train for 450 epochs with a batch size of 1024.

For ImageNet, we also used SGD momentum optimizer (Nesterov=False, momentum=0.9) with a linear learning rate ramp up for the first 5 epochs (peaked at 0.8) and decayed by a factor of 10 at epochs 30, 60 and 80. In total, we train for 90 epochs with a batch size of 1024.

**Hyperparameters**  We provide additional hyperparameter details for the experiments. (PG stands for Patch Gaussian):

- **MNIST**
  FCL $\sigma = 0.5, \tau = 0.1, \lambda = 0.001$
- **Noisy CIFAR-10**
  XE+Gaussian $\sigma = 0.3, \lambda = 0.0001$
  CL+Gaussian $\sigma = 0.5, \tau = 0.5, \lambda = 0.0001$, ramp_up=14000steps
  FCL $k = 256, \sigma = 0.5, \tau = 2, \lambda = 0.0001$, ramp_up=14000steps
  PG $\sigma = 0.1$, patch_size=25
  PG+FCL $k = 256, \sigma = 0.5, \tau = 1, \lambda = 0.0001$, ramp_up=14000steps (PG $\sigma = 0.1$, patch_size=25)
- **Noisy CIFAR-100**
  XE+Gaussian $\sigma = 0.3, \lambda = 0.0001$
  CL+Gaussian $\sigma = 0.5, \tau = 0.5, \lambda = 0.0001$, ramp_up=10000steps
  FCL $k = 256, \sigma = 0.5, \tau = 0.1, \lambda = 0.0001$, ramp_up=10000steps
  PG $\sigma = 0.05$, patch_size=25
  PG+FCL $k = 256, \sigma = 0.5, \tau = 0.1, \lambda = 0.0001$, ramp_up=10000steps (PG $\sigma = 0.05$, patch_size=25)

- **CIFAR-10-C**
  XE+Gaussian $\sigma = 0.3, \lambda = 0.0001$
  CL+Gaussian $\sigma = 0.5, \tau = 0.5, \lambda = 0.0001$, ramp_up=14000steps
  FCL $k = 256, \sigma = 0.5, \tau = 2, \lambda = 0.0001$, ramp_up=14000steps
  PG $\sigma = 0.1$, patch_size=25
  PG+FCL $k = 256, \sigma = 0.5, \tau = 1, \lambda = 0.0001$, ramp_up=10000steps (PG $\sigma = 0.1$, patch_size=25)

- **CIFAR-100-C**
  XE+Gaussian $\sigma = 0.3, \lambda = 0.0001$
  CL+Gaussian $\sigma = 0.5, \tau = 0.5, \lambda = 0.0001$, ramp_up=10000steps
  FCL $k = 256, \sigma = 0.5, \tau = 0.1, \lambda = 0.0001$, ramp_up=10000steps
  PG $\sigma = 0.1$, patch_size=25
  PG+FCL $k = 256, \sigma = 0.5, \tau = 0.1, \lambda = 0.0001$, ramp_up=10000steps (PG $\sigma = 0.05$, patch_size=25)

- **ImageNet-C**
  XE+Gaussian $\sigma = 0.5, \lambda = 0.0001$
  CL+Gaussian $\sigma = 0.5, \tau = 0.5, \lambda = 0.0001$, ramp_up=78000steps
  FCL $k = 512, \sigma = 1.0, \tau = 0.5, \lambda = 0.0002$
  PG $\sigma = 1.0$, patch_size $\leq 250$
  PG+FCL $k = 2048, \sigma = 0.5, \tau = 1.0, \lambda = 0.0004$, ramp_up=78000steps (PG $\sigma = 1.0$, patch_size $\geq 250$)

## C   FULL CIFAR-10-C, CIFAR-100-C AND IMAGENET-C ACCURACY

| Dataset | Method | Noise | | | | Blur | | |
|---|---|---|---|---|---|---|---|---|
| | | gauss. | shot | impulse | defocus | glass | motion | zoom |
| CIFAR-10-C | XE | 0.4049 | 0.5374 | 0.5477 | 0.7912 | 0.4846 | 0.7350 | 0.7222 |
| | XE+Gaussian | 0.5029 | 0.6221 | 0.6149 | 0.7992 | 0.5192 | 0.7315 | 0.7369 |
| | CL+Gaussian | 0.4767 | 0.5922 | 0.5650 | 0.7919 | 0.5383 | 0.7238 | 0.7215 |
| | FCL | 0.5505 | 0.6431 | 0.7311 | 0.7922 | 0.5140 | 0.7273 | 0.7210 |
| | PG+XE | **0.8995** | **0.9082** | 0.8776 | 0.8252 | 0.6731 | **0.7634** | 0.7883 |
| | PG+FCL | 0.8983 | 0.9078 | **0.8918** | 0.8268 | **0.6764** | 0.7601 | **0.7924** |
| CIFAR-100-C | XE | 0.1644 | 0.2446 | 0.2249 | 0.5577 | 0.2022 | 0.4884 | 0.4808 |
| | XE+Gaussian | 0.2024 | 0.2816 | 0.2667 | 0.5557 | 0.2055 | 0.4812 | 0.4807 |
| | CL+Gaussian | 0.1880 | 0.2668 | 0.2502 | 0.5562 | 0.2110 | **0.4903** | 0.4827 |
| | FCL | 0.2551 | 0.3186 | 0.4847 | 0.5571 | 0.2137 | 0.4861 | 0.4852 |
| | PG+XE | 0.1773 | 0.2542 | 0.2279 | **0.5596** | 0.2001 | 0.4897 | **0.4883** |
| | PG+FCL | **0.2729** | **0.3349** | **0.5020** | 0.5487 | **0.2310** | 0.4845 | 0.4768 |
| ImageNet-C | XE | 0.2860 | 0.2651 | 0.2335 | 0.2945 | 0.2312 | 0.2844 | 0.3163 |
| | XE+Gaussian | 0.2876 | 0.2654 | 0.2339 | 0.2978 | 0.2293 | 0.2851 | 0.3193 |
| | CL+Gaussian | 0.2898 | 0.2694 | 0.2383 | 0.2955 | 0.2290 | 0.2873 | 0.3177 |
| | FCL | 0.2954 | 0.2738 | 0.2395 | 0.2989 | 0.2374 | 0.2861 | 0.3176 |
| | PG+XE | 0.3265 | 0.3070 | **0.2822** | 0.3333 | **0.2571** | **0.2923** | **0.3337** |
| | PG+FCL | **0.3304** | **0.3107** | 0.2821 | **0.3344** | **0.2571** | 0.2920 | 0.3302 |

| Dataset | Method | Weather | | | | Digital | | | |
|---|---|---|---|---|---|---|---|---|---|
| | | snow | forest | fog | bright | contrast | elastic | pixel | JPEG |
| CIFAR-10-C | XE | 0.7933 | 0.7486 | 0.8587 | **0.9231** | **0.7310** | 0.8092 | 0.6991 | 0.7756 |
| | XE+Gaussian | 0.8027 | 0.7711 | **0.8601** | 0.9218 | 0.7240 | 0.8104 | 0.7195 | 0.7750 |
| | CL+Gaussian | 0.7930 | 0.7589 | 0.8542 | 0.9189 | 0.7267 | 0.8023 | 0.7083 | 0.7747 |
| | FCL | 0.7988 | 0.7555 | 0.8590 | 0.9217 | 0.7233 | 0.8092 | 0.7036 | 0.7758 |
| | PG+XE | 0.8306 | 0.8353 | 0.8303 | 0.9198 | 0.7085 | **0.8417** | 0.7919 | **0.8664** |
| | PG+FCL | **0.8324** | **0.8374** | 0.8256 | 0.9189 | 0.6937 | 0.8416 | **0.7994** | 0.8652 |
| CIFAR-100-C | XE | 0.5023 | 0.4370 | **0.5874** | 0.6842 | **0.4884** | 0.5458 | 0.4500 | 0.4578 |
| | XE+Gaussian | **0.5061** | **0.4403** | 0.5825 | 0.6805 | 0.4793 | 0.5431 | 0.4582 | 0.4585 |
| | CL+Gaussian | 0.5034 | 0.4392 | 0.5822 | 0.6807 | 0.4796 | **0.5484** | 0.4560 | 0.4621 |
| | FCL | 0.5007 | 0.4333 | 0.5748 | 0.6779 | 0.4640 | 0.5441 | 0.4583 | **0.4724** |
| | PG+XE | 0.4970 | 0.4308 | 0.5834 | 0.6819 | 0.4843 | 0.5483 | **0.4635** | 0.4623 |
| | PG+FCL | 0.5050 | **0.4403** | 0.5756 | 0.6750 | 0.4687 | 0.5394 | 0.4634 | 0.4688 |
| ImageNet-C | XE | 0.2773 | 0.3304 | 0.4695 | 0.6083 | 0.3273 | 0.4096 | 0.2998 | 0.4763 |
| | XE+Gaussian | 0.2748 | 0.3323 | 0.4736 | 0.6088 | 0.3298 | 0.4101 | 0.2989 | 0.4743 |
| | CL+Gaussian | 0.2745 | 0.3327 | 0.4728 | 0.6091 | 0.3309 | 0.4060 | 0.2978 | 0.4768 |
| | FCL | 0.2739 | 0.3297 | 0.4674 | 0.6044 | 0.3278 | 0.4143 | 0.3111 | 0.4777 |
| | PG+XE | **0.2891** | 0.3464 | 0.4735 | **0.6110** | 0.3352 | **0.4331** | 0.3232 | **0.4939** |
| | PG+FCL | 0.2880 | **0.3495** | **0.4761** | 0.6097 | **0.3368** | 0.4313 | **0.3290** | 0.4934 |

Table 4: Image classification accuracies on the CIFAR-10-C, CIFAR-100-C and ImageNet-C sets (Hendrycks & Dietterich, 2019). PG stands for Patch Gaussian data augmentation (Lopes et al., 2019). All FCL means $FCL_{xe}$. All numbers are averaged by 5 runs.

