# OpenReview forum: "Balancing Robustness and Sensitivity using Feature Contrastive Learning"
_ICLR.cc/2021/Conference — Reject_

### Official Review · AnonReviewer2 · 2020-10-24
**Review #2**

**Rating:** 5
**Confidence:** 4

**Review:**

Summary:
This paper introduces the concept of contextual feature utility and sensitivity to illustrate the trade-off between robustness and sensitivity. The authors propose Feature Contrastive Learning (FCL) to regularize models to be more sensitive to features that have higher utility, i.e. change the classification loss of the model to a larger extent. FCL first ranks features according to how much they change the loss values. Gaussian perturbations are then added to the top- and bottom-ranked pixels to form negative and positive pairs with the original input respectively. Contrastive learning is conducted by training the feature extractor of the classifier to minimize distance between the two hidden states in a particular positive pair, in order to align feature sensitivity with utility. The authors create a synthetic MNIST classification task where a subset of digit classes are affected by the presence of other digits at the corner of the image, to test a model’s selective sensitivity to input features. Through experiments on the synthetic MNIST and two variants of CIFAR datasets, FCL is shown to improve classification accuracy under noisy conditions while maintaining good clean accuracy.

Pros:

+Good direction to train models that strive to have a better balance between robustness against label-preserving perturbations and sensitivity towards label-changing perturbations.

+The idea of using contrastive learning to improve the robustness of models is interesting and could potentially be used for other kinds of perturbations.


Cons:

-Lack of studies on robustness against adversarial examples (L-p and invariance types), experiments currently only show results on robustness on gaussian/uniform noise.

-Little theoretical support or justification on why aligning the feature sensitivity and utility proposed here would help balance between sensitivity and robustness (such as against L-p and invariance adversarial examples).

Recommendation:
While the idea of training a model to be both robust and sensitive at the same time is well-motivated and promising, the experiments here fall short of evaluating robustness beyond gaussian/uniform noise. The claims of the paper would be stronger with robustness evaluation against the widely-studied L-p norm and recent invariance adversarial examples that are also mentioned in this paper.

Considering the lack of the aforementioned experiments and of more rigorous theoretical support for FCL for robustness that generalizes beyond the gaussian/uniform noise evaluated here, this paper is still not ready for publication.

Other questions and comments:
What is the performance of the model if Contrastic Learning is done by creating positive pairs by just adding gaussian noise to the original image?

Since the proposed method, FCL, relies on contrastive learning, it would help to discuss prior work on contrastive learning, especially highly similar ones such as: https://arxiv.org/pdf/2006.07589.pdf

---

> ### Author Response · Authors · 2020-11-17
> **Adversarial robustness versus natural perturbation robustness**
>
> We thank the reviewer for their thoughtful review and valuable feedback!
>
> Regarding adversarial perturbations, we view adversarial robustness and robustness to so-called common or natural perturbations as two different tasks, both of which are important. In this paper our focus is on the latter. In fact, our goal of “making models sensitive to important features” implies that the model should not be adversarially robust on high utility features.
>
> Adversarial robustness aims to protect from adversarially-generated small magnitude perturbations, whereas robustness to natural perturbations addresses large magnitude perturbations drawn from specific “natural” distributions such as sensor corruptions or blur. To validate that our method improves robustness to natural perturbations, we conduct experiments on CIFAR-10 and CIFAR-100 with synthetic noise injection and 19 corrupted patterns introduced in [Hendrycks & Dietterich, 2019], which is a standard benchmark for evaluating robustness to natural perturbations.
>
> Recent papers [Laugros et al., 2019, Gulshad et al., 2020] discuss the relationship between adversarial robustness and natural perturbation robustness, and find that they are usually poorly correlated. For example, Table 5 in [Laugros et al., 2019] shows models trained for adversarial robustness that are not more robust than standard models on common perturbation benchmarks. The converse is also shown (Section 4.2 in [Laugros et al., 2019]). [Gulshad et al., 2020] also found a similar trend.
>
> While both are important goals, we highlight some key differences between these tasks.
> * The average perturbation magnitude in the datasets used in our experiments is much larger than the perturbation radii typically used in the adversarial robustness setting. For example the authors of [Yang et al. 2020] cite 0.031 as a typical perturbation radius for CIFAR-10 in the adversarial robustness literature. We measured the average perturbation magnitudes in CIFAR-10-C, and find that they are larger by an order of magnitude. CIFAR-10-C includes 5 levels of severity, the average $\ell_{\infty}$ norm over all perturbations in the lowest severity level is 0.259, and 0.453 for the highest.
> * Adversarial robustness usually comes at the cost of significant negative impact on the clean accuracy [Zhang et al., 2019, Mandry et al., 2017]; however, FCL does not hurt the clean accuracy.
>
> Regarding [Kim et al., 2020]: We thank the reviewer for pointing out this highly relevant paper! We will be happy to add a discussion of this paper in ours. [Kim et al., 2020] and ours differ in three main aspects. a) Their paper focuses on adversarial robustness while ours focuses on robustness to natural perturbations. b) Since their paper focuses on the adversarial robustness, their contrastive learning effectively suppresses the distance between the original and an adversarially perturbed image (against random pairs). In contrast, we apply contrastive learning to low- and high-utility perturbation pairs, encouraging the former to be close and the latter to be far. c) [Kim et al., 2020]’s perturbation is based on an unsupervised loss (instance identification task), while ours rely on class labels to identify feature dimensions that are useful/not useful for the main classification task.
>
> Regarding contrastive learning with Gaussian noise: We will run the experiment and update as soon as we have results. We expect this new baseline to behave similarly to our “Gaussian” baseline in Section 4.2. Both methods encourage the original image and a version with added Gaussian noise to be close in the embedding space.
>
> References:
> * [Gulshad et al., 2020] Gulshad, Sadaf, Jan Hendrik Metzen, and Arnold Smeulders. "Adversarial and Natural Perturbations for General Robustness." arXiv e-prints (2020): arXiv-2010.
> * [Kim et al., 2020] Kim, Minseon, Jihoon Tack, and Sung Ju Hwang. "Adversarial self-supervised contrastive learning." Advances in Neural Information Processing Systems 33 (2020).
> * [Laugros et al., 2019] Laugros, Alfred, Alice Caplier, and Matthieu Ospici. "Are Adversarial Robustness and Common Perturbation Robustness Independent Attributes?." Proceedings of the IEEE International Conference on Computer Vision Workshops. 2019.
> * [Mandry et al., 2017] Madry, Aleksander, et al. "Towards deep learning models resistant to adversarial attacks." arXiv preprint arXiv:1706.06083.
> * [Yang et al., 2020] Yang, Yao-Yuan, et al. "A closer look at accuracy vs. robustness." Advances in Neural Information Processing Systems 33 (2020).
> * [Zhang et al., 2019] Zhang, Hongyang, et al. "Theoretically principled trade-off between robustness and accuracy." arXiv preprint arXiv:1901.08573 (2019).

---

> ### Author Response · Authors · 2020-11-24
> **Re: Contrastive learning with the gaussian noise as the positive pair**
>
> Per your request, we added a new baseline “CL+Gaussian” that creates the positive pair by adding Gaussian noise to the original image instead of FCL’s utility-dependent perturbation. We evaluated the new baseline on all (larger-scale) experiments and updated (Table 2, Table 3 and Appendix C) with the results. We also updated the relevant discussions in (Section 4.2).
>
> Thank you.

---

### Official Review · AnonReviewer1 · 2020-10-28
**a new contrastive learning based methods to obtain stable features**

**Rating:** 6
**Confidence:** 3

**Review:**

Summary of work
The authors introduce the concept of contextual sensitivity to describe the importance of the feature, which is defined as the absolute value of the Jacobian of loss with respect to the input. High-utility and low-utility perturbations are created by perturbing most important and least important input variables respectively. The embedding of the original input forms a positive pair with the embedding of high-utility perturbation, and forms a negative pair with the embedding of low-utility perturbation, based on which two contrastive loss functions are proposed.

strength :
The authors propose a novel feature perturbation approach by performing perturbation to features with low or high sensitivity distinctively, based on which a contrastive learning loss is developed. Experiments are conducted on CIFAR10, CIFAR100 dataset, and a new synthetic MNIST dataset. The performance of the proposed approach surpasses baseline methods.

weakness :
The authors define two loss functions to calculate the contrastive loss and choose the latter one for all the experiments. It would be nice if the authors could provide an explanation for such preference. Does the latter loss function perform better than the previous one? How the data attribute affect the choice of different contrastive loss?

---

> ### Author Response · Authors · 2020-11-17
> **Clarifying different contrastive loss functions**
>
> We thank the reviewer for their thoughtful review and valuable feedback!
>
> FCL is compatible with different contrastive loss functions, and part of our motivation to introduce two formulations was to demonstrate this point. In particular, FCL_margin explicitly encourages embeddings of low-utility perturbations to be as close as possible to the original version, and those corresponding to high utility perturbations to be separated by at least a margin. FCL_XE on the other hand, is applied to all pairs in the batch and imposes ranking constraints rather than an explicit separation by a margin.
>
> Empirically, we tried both loss functions in the MNIST experiments and found that they achieve  similar performance. We chose FCL_XE for the other experiments since it doesn’t require tuning the margin parameter.

---

### Official Review · AnonReviewer4 · 2020-10-28
**Interesting and simple idea**

**Rating:** 7
**Confidence:** 4

**Review:**

### Summary
In this work, the authors focus on the robustness against only common corruptions and perturbations by defining a contextual feature utility metric. It measures the magnitude of the change in the loss of a perfect model that an input feature can incur. They leverage this metric to design a utility-aware perturbation that they use to control the trade-off between model robustness and its sensitivity to high utility features. They formulate the problem as contrastive loss which can be added as a regularizer for advanced training stages. They dubbed this method as Feature Contrastive Learning (FCL). Finally, they defined another metric dubbed contextual feature sensitivity that is loosely defined as the magnitude of the change in the activations of a model that an input feature can incur.

### Strengths
1. FCL is a simple method that can be added as a fine-tuning step
1. Clear submission with sound experimental setup and sufficient results on small synthetic and real datasets

### Weaknesses
1. The impact of this work would greatly benefit from ImageNet experiments as it is the main benchmark for this limited type of robustness
1. Two metrics were introduced (utility and sensitivity) but only one of them was used in the rest of the work with only a hand-wavy explanation of their relationship (e.g., it would be interesting to see how they interplay when training with and without FCL)
1. The motivation behind the contextual feature utility metric should make it model-independent but defined as the model's own loss gradient w.r.t. the input (e.g. in practice, it could be defined by a pre-trained model instead and used for training)

Can the authors comment on the negative points mentioned above?

### Rating
I like the simplicity of the idea and its applicability. However, I gave it this score mainly because of the previous reasons and its potential impact. Since the authors are interested in these simple corruptions, the value of the work is hindered by the absence of ImageNet experiments.

### Verdict
Thank you for addressing all my concerns. This gives me the confidence to slightly increase my score.

One more small thing, for future work, you might also consider incorporating clipping the metrics to the input range.

---

> ### Author Response · Authors · 2020-11-17
> **Clarifying sensitivity/utility usage, Imagenet training, and pre-trained utility measures**
>
> We thank the reviewer for their thoughtful review and valuable feedback!
> * Thanks for suggesting ImageNet experiments. We are running experiments, and we will update as soon as we have the results.
> * Regarding “two metrics”: thank you for drawing our attention to a part of the text which might be confusing, we will revise the paper to clarify.
>
>   Our feature contrastive learning uses both contextual utility and contextual sensitivity simultaneously. The utility is used for selecting the features, and the associated sensitivity values are adjusted by applying the contrastive loss. In particular, the utility identifies the features we want to perturb, and our loss function tries to increase or decrease the sensitivity of the model to those features, based on their utility values.
>
>   To provide further intuition for the connection between utility and sensitivity, consider a classification task with cross entropy loss, and let $f()$ be the output of the network after applying a softmax. In this setting, the loss is minus log probability of the correct label.
>
>   The utility of $f()$ is defined by
>   $$
> u = \left| \frac{ \partial \ell(y, f(x; w)) } { \partial x}  \right| = \left| \frac{ \partial \log[ f(x; w)_{y} ] } { \partial x } \right| = \frac{1}{ {f(x; w)}_y }  \left| \frac{\partial {f(x; w)}_y } { \partial x }  \right|
>  $$
>   Also recall that the sensitivity of $f()$ is given by
>   $$
> s = \left| \left| \frac{\partial f(x; w)}{\partial x} \right| \right|  = \sqrt{\sum_c \frac{\partial f(x; w)_c^2}{\partial x}}.
>   $$
>   We can see that the utility is a product of two terms. The first is the reciprocal of the networks’ prediction for the correct class, and the second is the sensitivity term specific to the correct class. When the network’s prediction is correct the utility is proportional to the ground truth class’s sensitivity. If changing the feature will not affect the correct prediction it doesn’t have much utility and vice versa. On the other hand, when the network makes a mistake, the utility will be large regardless of the ground truth class’s sensitivity. Our algorithm takes advantage of this behavior to promote robustness and maintain sensitivity.
> * Thanks for suggesting that we can use a “a pre-trained model” for the utility function. This is an excellent idea! Indeed, FCL can be applied with an “oracle utility” or a “teacher utility” (as in distillation) providing the model with guidance on contextual utility. For example we could consider a human, or a more powerful pre-trained model. We hope to explore this idea in future work.

---

> ### Author Response · Authors · 2020-11-24
> **ImageNet-C experiments**
>
> We added the requested ImageNet-C experiment on the updated version of the paper. The results are reported in (Table 3 and Appendix C). We updated discussion of the results in (Section 4.2) and experiment details in (Section 4.2 and Appendix B).
>
> Thank you.

---

### Official Review · AnonReviewer3 · 2020-10-30
**Official Blind Review # 3**

**Rating:** 5
**Confidence:** 2

**Review:**

Summary & Pros:
- This paper introduces contextual feature utility and contextual feature sensitivity to measure and identify high utility features and their associated model sensitivity, and proposes Feature Contrastive Learning to  balance robustness
and sensitivity in deep neural network training.
- For the evaluation, the analysis experiments are extensive.

However, I have still some concerns below:
- Topic Concerns. The goal of this work is to balance robustness and sensitivity. In fact, I am confused the definition of robustness and sensitity as adversarial robustness contains the concept of sensitity.
- There is not much related work in the paper , and I don't know how important the direction is.

---

> ### Author Response · Authors · 2020-11-17
> **Clarifying robustness/sensitivity definitions and the emphasizing problem impact**
>
> We thank the reviewer for their thoughtful review and valuable feedback!
> We address your concerns below:
> * Regarding the “topic concern”, we apologize for any misunderstanding and we will clarify this better in the paper. Yes, robustness and sensitivity are overloaded terms and used in many different contexts. To clarify, we use the term “robustness” as the generic notion of model stability -- how resilient the model prediction would be when we change the input. The sensitivity is simply the opposite meaning in this context -- how much a sensitive model changes their prediction when we change the input. We view adversarial robustness and robustness to so-called common or natural perturbations as two different goals. In this paper our focus is on the latter. (Please see our response titled “Adversarial robustness vs. natural perturbation robustness” for additional details.)
> * We completely agree. There are not many papers on this topic. We believe that this is an important problem and that both robustness and sensitivity should be targeted in model training. While robustness (especially the adversarial case) has received much attention, the importance of balancing these measures has received very little attention. A few recent papers [Tsipras et al., 2018, Schmidt et al., 2018, Zhang et al., 2019, Tramèr et al., 2020, Yang et al., 2020] have started emphasizing the importance of this problem, but do not provide a solution. We propose a method to balance both sensitivity and robustness.
>
> References
> * [Schmidt et al., 2018] Ludwig Schmidt, Shibani Santurkar, Dimitris Tsipras, Kunal Talwar, and Aleksander Madry. Adversarially robust generalization requires more data. In Advances in Neural Information Processing Systems, pp. 5014–5026, 2018.
> * [Tramèr et al., 2020] Tramèr, Florian, et al. "Fundamental tradeoffs between invariance and sensitivity to adversarial perturbations." arXiv preprint arXiv:2002.04599, 2020
> * [Tsipras et al., 2018] Dimitris Tsipras, Shibani Santurkar, Logan Engstrom, Alexander Turner, and Aleksander Madry.
> * [Zhang et al., 2019] Zhang, Hongyang, et al. "Theoretically principled trade-off between robustness and accuracy." arXiv preprint arXiv:1901.08573 (2019).
> * [Yang et al., 2020] Yang, Yao-Yuan, et al. "A closer look at accuracy vs. robustness." Advances in Neural Information Processing Systems 33 (2020).

---

### Comment · Area_Chair1 · 2020-11-18
**Authors: Thank you for response / Reviewers: Please update**

Thank you, authors, for your responses.

Reviewers, please read the responses and update your reviews by stating that your concerns have been addressed or by providing further rebuttal.

---

### Author Response · Authors · 2020-11-24
**Paper update**

We thank the reviewers for their valuable comments! We’ve updated the paper to address them.

The main changes are as follows:
 * **Adversarial robustness vs. natural perturbation robustness:** We clarified the distinction between adversarial robustness and natural perturbation robustness, and added literature relevant to the distinction (Section 1). We also clarified the contribution of our work in this context (Section 1).
 * **Sensitivity and utility:** We clarified how FCL uses both contextual feature sensitivity and utility and the interplay between them (Section 3). We also added a more detailed discussion of their relationship in (Appendix A).
 * **Loss functions:** We clarified the use of different contrastive loss functions in FCL (Section 3).
 * **New contrastive learning baseline:** Per AnonReviewer2’s request, we added a new baseline “CL+Gaussian” that creates the positive pair by adding Gaussian noise to the original image instead of FCL’s utility-dependent perturbation. We evaluated the new baseline on all (larger-scale) experiments and updated (Table 2, Table 3 and Appendix C) with the results. We also updated the relevant discussions in (Section 4.2).
 * **ImageNet-C:** Addressing AnonReviewer4’s request, we added an experiment evaluating all methods on ImageNet-C [Hendrycks et al., 2019]. The results are reported in (Table 3 and Appendix C). We updated discussion of the results in (Section 4.2) and experiment details in (Section 4.2 and Appendix B).

Thank you for your time, and we hope that the changes address all of your concerns.

[Hendrycks et al., 2019] Hendrycks, Dan, and Thomas Dietterich. "Benchmarking neural network robustness to common corruptions and perturbations." arXiv preprint arXiv:1903.12261 (2019)]

---

### Decision · Program_Chairs · 2021-01-07
**Final Decision**

**Decision:**

Reject

**Comment:**

This paper proposes Feature Contractive Learning (FCL), a training framework that takes a more nuanced view of robustness, refining it to the sensitivity of the feature.  There are some differing opinions among the reviewers, with some applauding the simplicity of this new take on robustness while others are unsure of its underlying definitions and relationship to adversarial robustness.  The authors claimed to have clarified some of these points in their rebuttal / revision, but unfortunately, there was not much follow-up discussion by the reviewers.  Ultimately, there are still enough lingering issues that rejection is warranted.